# A Progressive Training Framework for Spiking Neural Networks with Learnable Multi-hierarchical Model

**Zecheng Hao[1], Xinyu Shi[1,2], Zihan Huang[1], Tong Bu[1,2], Zhaofei Yu[1,2]\* & Tiejun Huang[1,2]**
[1] School of Computer Science, Peking University
[2] Institute for Artificial Intelligence, Peking University

## Abstract

Spiking Neural Networks (SNNs) have garnered considerable attention due to their energy efficiency and unique biological characteristics. However, the widely adopted Leaky Integrate-and-Fire (LIF) model, as the mainstream neuron model in current SNN research, has been revealed to exhibit significant deficiencies in deep-layer gradient calculation and capturing global information on the time dimension. In this paper, we propose the Learnable Multi-hierarchical (LM-H) model to address these issues by dynamically regulating its membrane-related factors. We point out that the LM-H model fully encompasses the information representation range of the LIF model while offering the flexibility to adjust the extraction ratio between historical and current information. Additionally, we theoretically demonstrate the effectiveness of the LM-H model and the functionality of its internal parameters, and propose a progressive training algorithm tailored specifically for the LM-H model. Furthermore, we devise an efficient training framework for our novel advanced model, encompassing hybrid training and time-slicing online training. Through extensive experiments on various datasets, we validate the remarkable superiority of our model and training algorithm compared to previous state-of-the-art approaches. Code is available at https://github.com/hzc1208/STBP_LMH.

## 1 Introduction

Spiking Neural Networks (SNNs) are considered as the third generation of artificial neural networks (Maass, 1997), which represents a fascinating paradigm with superior biological plausibility and high energy efficiency in the field of computational neuroscience. Unlike traditional Analog Neural Networks (ANNs), which rely on continuous-valued outputs, SNNs draw inspiration from the structure of the human brain and emulate the behavior of real biological neurons by utilizing binary and sparse spiking signals to transmit information. In SNNs, neurons only generate spikes and transmit them to the postsynaptic layer when their membrane potential exceeds a certain firing threshold. This event-driven characteristic of SNNs provides significant advantages in terms of computational efficiency and power consumption reduction (Roy et al., 2019), especially when deployed on neuromorphic chips (Merolla et al., 2014; Davies et al., 2018; DeBole et al., 2019). In recent years, SNNs have garnered increasing attention due to their potential to address certain limitations of ANNs. They have shown promise in applications such as image recognition (Fang et al., 2021b), object detection (Kim et al., 2020b), natural language processing (Lv et al., 2023), cue combination (Yu et al., 2020), and network robustness (Bu et al., 2023), etc.

The Leaky Integrate-and-Fire (LIF) model (Gerstner & Kistler, 2002) is a widely used spiking neuron model that consists of three main processes: charging, membrane potential leakage, and firing of spikes. However, the non-differentiable nature of the spike firing process poses challenges for gradient calculations. To address this issue, researchers have developed the current mainstream Spatio-Temporal Back-propagation (STBP) algorithm (Wu et al., 2018), which introduces surrogate gradients to enable gradient-based learning in LIF-based SNNs. Nevertheless, the LIF model still

---

\*Corresponding author: yuzf12@pku.edu.cn

encounters gradient calculation failures when applied to deep residual networks (Fang et al., 2021a). Additionally, the LIF model struggles to effectively utilize historical information, such as the input current and membrane potential at previous time-steps, which can be crucial for capturing temporal dependencies and improving network performance.

To address these issues, we propose a novel Learnable Multi-hierarchical (LM-H) model. In the LM-H model, the scaling factors from the dendrite layer regulate the proportion of historical information extracted by the model, while the factors from the soma layer determine the degree of potential leakage and the intensity of the input current at present. By effectively leveraging these factors, the LM-H model enables precise gradient calculation and optimal learning performance. It is worth noting that our model is inspired by the TC-LIF model (Zhang et al., 2023). However, the TC-LIF model suffers from the issue of setting certain membrane-related parameters to fixed or contradictory values. Moreover, it fails to accurately recognize the relationship between the LIF and advanced neurons, resulting in a significant impact on the stability of learning gradients. In this work, we investigate the relationship between the LIF and LM-H models, and achieve more precise information extraction and gradient propagation by designing appropriate optimization intervals and calculation modes for membrane-related parameters. Our main contributions can be summarized as follows:

- We identify the limitations of the vanilla LIF model in terms of its representation capabilities and propose the LM-H model with a wider calculation scope. We mathematically demonstrate that our proposed model can effectively extracting global information along the time dimension and propagate gradients in deep networks.
- We systematically analyze the specific roles of parameters on the dendrite and soma layers, and further develop a progressive STBP training algorithm for the LM-H model, which can dynamically optimize the membrane-related parameters during the learning process.
- To enhance the energy efficiency of SNN learning, we propose an efficient training framework specifically designed for the LM-H model, which includes hybrid training and time-slicing online training.
- Experimental results validate the significant advantages of the LM-H model in the field of SNN supervised learning. Our proposed method achieves state-of-the-art performance on multiple datasets with various scales and data types. For example, we achieve $80.31\%$ top-1 accuracy with 4 time-steps on CIFAR-100 using the ResNet-19 architecture.

## 2 RELATED WORK

**Supervised Learning for SNNs.** STBP algorithm has emerged as the most mainstream supervised learning approach in the SNN community, which draws inspiration from the Back-Propagation Through Time (BPTT) gradient computation framework used in Recurrent Neural Networks (RNNs). Pioneered by researchers such as Wu et al. (2018) and Neftci et al. (2019), STBP addresses the non-differentiable nature of firing spikes during the back-propagation process by employing surrogate gradient functions. Building upon the foundation laid by STBP, subsequent studies have further enhanced the accuracy and stability of SNN training. For instance, Li et al. (2021) and Wang et al. (2023) proposed families of differentiable functions that can be adaptively evolved during the learning process, leading to improved training outcomes. Deng et al. (2022) and Guo et al. (2022a) introduced novel loss functions to regulate the distribution of spikes and membrane potentials along the temporal dimension, resulting in more precise alignment of learning gradients. Furthermore, researchers have developed various batch normalization structures tailored for SNN learning (Wu et al., 2019; Zheng et al., 2021; Duan et al., 2022). These structures facilitate better network performance and training convergence. Additionally, Fang et al. (2021a) significantly advanced the effective learning of SNNs on deep networks with more than 100 layers by improving the connection patterns of residual blocks. It is worth noting that supervised learning methods based on spike firing timing (Mostafa, 2017; Kim et al., 2020a; Zhou et al., 2021) have garnered significant attention in recent years. However, these approaches are currently limited to shallow networks and small-scale datasets, and further research is needed to extend their applicability to more complex scenarios (Zhu et al., 2022).

**Efficient Training for SNNs.** The ANN-SNN Conversion (Cao et al., 2015; Han et al., 2020; Bu et al., 2022) is another widely used algorithm family of SNN learning, which converts pre-trained ANNs into SNNs based on the approximate linear relationship between the average spike firing rate of adjacent layers. While the precision of ANN-SNN Conversion may be compromised under a few time-steps (Hao et al., 2023a;b), it offers significant time and memory savings compared to

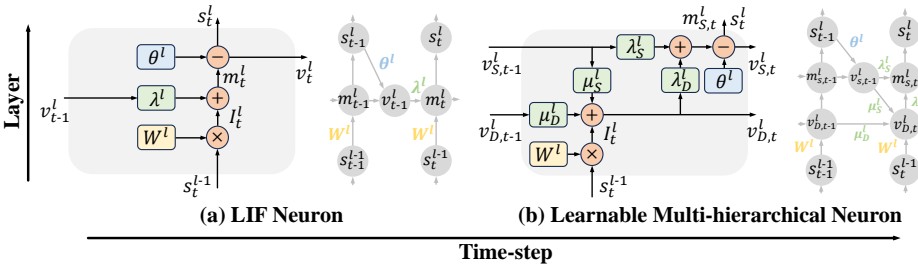

Figure 1: The structural description for the vanilla LIF and LM-H models.

STBP. Furthermore, efficient training frameworks such as online learning (Xiao et al., 2022; Yang et al., 2022; Meng et al., 2023) have been proposed. These frameworks perform gradient updates at each time-step, maintaining constant memory consumption instead of linearly increasing it. Hybrid training (Rathi & Roy, 2023; Wang et al., 2022) is another popular training method, which typically employs STBP to optimize and fine-tune parameters based on models obtained from ANN-SNN Conversion. Researchers have discovered that applying STBP for a limited number of epochs ($\leq 30$) can significantly enhance the performance of SNNs under low time-latency.

**Structural optimization for spiking neurons.** In early research, the relevant model parameters of spiking neurons were often viewed as hyper-parameters, which hindered the ability of SNNs to effectively capture and extract information during the training process. Rathi & Roy (2023) and Fang et al. (2021b) set the membrane leaky constant and firing threshold as learnable variables, thereby enhancing the biological similarity of spiking neurons. Stöckl & Maass (2021) further optimized membrane-related parameters by aiming to reduce spike firing rate. Another advanced model, GLIF (Yao et al., 2022), selectively regulates the input current, decay factor, and reset mechanism, thereby improving the dynamic learning ability of SNNs. Additionally, Wang et al. (2023) proposed a novel spiking neuron with mixed output to alleviate the impact of the non-differentiable problem on gradient precision. TC-LIF (Zhang et al., 2023) aimed to enhance the capability of SNNs to tackle long-term sequences, which also provided inspiration for this work. However, the problem of gradient divergence still exists for TC-LIF within a certain parameter range.

## 3 PRELIMINARIES

### 3.1 LEAKY INTEGRATE-AND-FIRE (LIF) NEURON MODEL

The LIF model is a widely used spiking neuron model in the neuroscience community. As shown in Fig.1(a), the LIF model emulates the functioning of biological neurons by incorporating three key processes: charging, leakage, and firing. The following equations describe the iterative version of the LIF model in discrete form.

$$\boldsymbol{m}^l[t] = \lambda^l \boldsymbol{v}^l[t-1] + \boldsymbol{I}^l[t], \quad \boldsymbol{v}^l[t] = \boldsymbol{m}^l[t] - \boldsymbol{s}^l[t]\theta^l.$$

$$\boldsymbol{I}^l[t] = \boldsymbol{W}^l \boldsymbol{s}^{l-1}[t]\theta^{l-1}, \quad \boldsymbol{s}^l[t] = H(\boldsymbol{m}^l[t] - \theta^l) = \begin{cases} 1, & \boldsymbol{m}^l[t] \geq \theta^l \\ 0, & \text{otherwise} \end{cases}. \tag{1}$$

In the equations, $\boldsymbol{m}^l[t]$ and $\boldsymbol{v}^l[t]$ represent the membrane potential before and after emitting a spike at the $t$-th time-step, respectively. $\boldsymbol{I}^l[t]$ denotes the presynaptic input current and $\boldsymbol{W}^l$ is the weight matrix of the $l$-th layer. The membrane decay constant $\lambda^l$ controls the extent to which neurons retain previous information. When $\lambda^l = 1$, the LIF model degenerates into the Integrate-and-Fire (IF) model. The binary variable $\boldsymbol{s}^l[t]$ determines whether a spike is fired or not. We choose the soft-reset mechanism in Eq.(1), where the membrane potential $\boldsymbol{v}^l[t]$ is decreased by a threshold magnitude when it exceeds the firing threshold $\theta^l$. The relationship between $\boldsymbol{s}^l[t]$ and $\boldsymbol{m}^l[t]$ is mathematically represented by the Heaviside step function $H(\cdot)$.

### 3.2 SPATIAL-TEMPORAL BACK-PROPAGATION (STBP) WITH SURROGATE GRADIENT

Drawing inspiration from the calculation mode of vanilla BPTT, we can transfer it to the learning of SNNs. As shown in Eq.(2), it can be observed that the gradient propagation in SNNs unfolds from

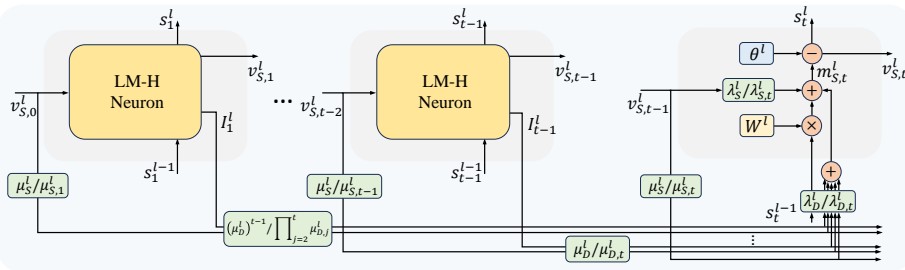

Figure 2: The specific extraction process of the historical and current information in the LM-H model.

back to front in both spatial and temporal dimensions. Here $\mathcal{L}$ represent the chosen loss function. Note that the Heaviside function is non-differentiable, which introduces an intractable node $\partial \boldsymbol{s}^l[t]/\partial \boldsymbol{m}^l[t]$ in the gradient calculation graph. To address it, previous works have used a surrogate function $H'(\cdot)$ that closely resembles the Heaviside function but is differentiable for back-propagation. In Eq.(3), we employ the well-known triangle function, where $\gamma$ controls the width of the non-zero interval.

$$\frac{\partial \mathcal{L}}{\partial \boldsymbol{m}^l[t]} = \frac{\partial \mathcal{L}}{\partial \boldsymbol{s}^l[t]} \frac{\partial \boldsymbol{s}^l[t]}{\partial \boldsymbol{m}^l[t]} + \frac{\partial \mathcal{L}}{\partial \boldsymbol{m}^l[t+1]} \frac{\partial \boldsymbol{m}^l[t+1]}{\partial \boldsymbol{m}^l[t]}, \tag{2}$$

$$\frac{\partial \boldsymbol{s}^l[t]}{\partial \boldsymbol{m}^l[t]} = H'(\boldsymbol{m}^l[t] - \theta^l) = \frac{1}{\gamma^2} \max\left(0, \gamma - |\boldsymbol{m}^l[t] - \theta^l|\right). \tag{3}$$

## 4 METHODOLOGY

### 4.1 REPRESENTATION DEFECTS OF THE LIF MODEL

**The gradient vanishing & exploding problem in deep residual architectures.** For the ResNet family, researchers have identified a potential issue with gradient calculation in the LIF model when it comes to the identity mapping path (Fang et al., 2021a). Considering a residual network with $L$ blocks, if we assume $\forall l \in [1, L], \boldsymbol{v}^l[t-1] = 0$ and choose an identity mapping path for each residual block, at the $t$-th time-step, we obtain the overall back-propagation chain $\prod_{l=1}^{L} \frac{\partial \boldsymbol{s}^l[t]}{\partial \boldsymbol{I}^l[t]} \frac{\partial \boldsymbol{I}^l[t]}{\partial \boldsymbol{s}^{l-1}[t]} = \prod_{l=1}^{L} H'(\boldsymbol{s}^{l-1}[t] - \theta^l)$. Since $\boldsymbol{s}^l[t]$ is a binary variable, the value of $H'(\boldsymbol{s}^{l-1}[t] - \theta^l)$ can be less or greater than 1. Consequently, as the number of blocks increases, there is a possibility that $\prod_{l=1}^{L} H'(\boldsymbol{s}^{l-1}[t] - \theta^l) \to 0$ or $\prod_{l=1}^{L} H'(\boldsymbol{s}^{l-1}[t] - \theta^l) \to +\infty$, leading to the ineffectiveness of identity mapping and gradient calculation.

**The inability to differentiate the current response through extracting past information.** Considering two LIF neurons $i$ and $j$ in the $l$-th layer, if $\boldsymbol{v}_i^l[t-1] = \boldsymbol{v}_j^l[t-1]$ and $\boldsymbol{I}_i^l[t] = \boldsymbol{I}_j^l[t]$, both neurons $i$ and $j$ will exhibit identical spike response and membrane potential renewal at the $t$-th step. However, during the preceding $t-1$ steps, neurons $i$ and $j$ may actually have different spike firing sequences $\{\boldsymbol{s}_i^l[1], ..., \boldsymbol{s}_i^l[t-1]\}$ and $\{\boldsymbol{s}_j^l[1], ..., \boldsymbol{s}_j^l[t-1]\}$, which corresponds to different latent information. Unfortunately, the LIF model lacks the capability to differentiate between neurons $i$ and $j$ at the $t$-th time step based on their previous spike firing patterns.

### 4.2 LEARNABLE MULTI-HIERARCHICAL (LM-H) NEURON MODEL

To overcome the limitations of the LIF model discussed earlier, we propose a brand-new LM-H neuron model. As shown in Fig.1(b), the LM-H model incorporates separate dendrite and soma layers. The input current flows through the dendrite layer before reaching the soma layer, where the decision to generate spikes is determined by the specific membrane potential of the soma layer. We

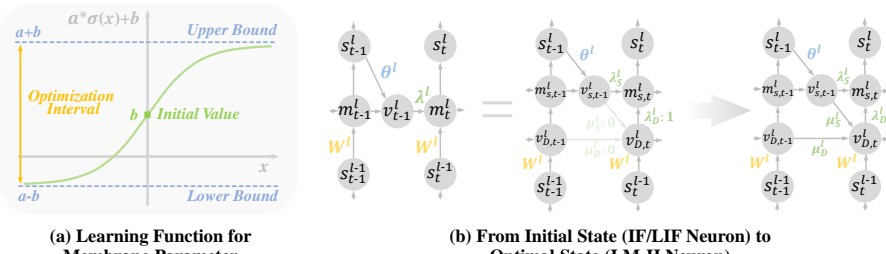

**(a) Learning Function for Membrane Parameter**

**(b) From Initial State (IF/LIF Neuron) to Optimal State (LM-H Neuron)**

Figure 3: Progressive STBP Training framework for the LM-H Neuron.

provide a dynamic description of the LM-H model as follows.

$$\boldsymbol{v}_D^l[t] = \mu_D^l \boldsymbol{v}_D^l[t-1] + \mu_S^l \boldsymbol{v}_S^l[t-1] + \boldsymbol{I}^l[t].$$

$$\boldsymbol{v}_S^l[t] = \boldsymbol{m}_S^l[t] - \boldsymbol{s}^l[t]\theta^l, \quad \boldsymbol{m}_S^l[t] = \lambda_S^l \boldsymbol{v}_S^l[t-1] + \lambda_D^l \boldsymbol{v}_D^l[t].$$

$$\boldsymbol{I}^l[t] = \boldsymbol{W}^l \boldsymbol{s}^{l-1}[t]\theta^{l-1}, \quad \boldsymbol{s}^l[t] = H(\boldsymbol{m}_S^l[t] - \theta^l) = \begin{cases} 1, & \text{if } \boldsymbol{m}_S^l[t] \geq \theta^l \\ 0, & \text{otherwise} \end{cases} \quad (4)$$

Here $\boldsymbol{v}_D^l[t]$ is the dendrite membrane potential at the $t$-th step. $\boldsymbol{m}_S^l[t]$ and $\boldsymbol{v}_S^l[t]$ denote the soma potential before and after firing a spike, respectively. $\mu_D^l, \mu_S^l, \lambda_D^l, \lambda_S^l$ are the relevant scaling factors of the LM-H model.

**Interpreting the variables and parameters in the LM-H model.** In order to facilitate a comprehensive analysis of this multi-layer neuron, we can transform the LM-H model into a single-layer form by applying Eq.(4) iteratively along the temporal dimension. This allows us to express all variables in terms of equations involving $\boldsymbol{v}_S^l[t]$ and $\boldsymbol{I}^l[t]$. Here we set $\boldsymbol{v}_D^l[0] = 0$ to simplify the expression.

$$\boldsymbol{v}_D^l[t] = \mu_S^l \boldsymbol{v}_S^l[t-1] + \boldsymbol{I}^l[t] + \mu_D^l \boldsymbol{v}_D^l[t-1]$$

$$= (\mu_S^l \boldsymbol{v}_S^l[t-1] + \boldsymbol{I}^l[t]) + \mu_D^l(\mu_S^l \boldsymbol{v}_S^l[t-2] + \boldsymbol{I}^l[t-1]) + (\mu_D^l)^2 \boldsymbol{v}_D^l[t-2]$$

$$= \sum_{k=1}^{t} (\mu_D^l)^{t-k}(\mu_S^l \boldsymbol{v}_S^l[k-1] + \boldsymbol{I}^l[k]), \quad (5)$$

$$\boldsymbol{m}_S^l[t] = \lambda_S^l \boldsymbol{v}_S^l[t-1] + \lambda_D^l \boldsymbol{v}_D^l[t]$$

$$= \underbrace{\sum_{k=1}^{t-1} \lambda_D^l(\mu_D^l)^{t-k}(\mu_S^l \boldsymbol{v}_S^l[k-1] + \boldsymbol{I}^l[k])}_{\text{historical representation}} + \underbrace{\left((\lambda_S^l + \lambda_D^l \mu_S^l)\boldsymbol{v}_S^l[t-1] + \lambda_D^l \boldsymbol{I}^l[t]\right)}_{\text{current representation}}. \quad (6)$$

In Eq.(5), one can find that $\boldsymbol{v}_D^l[t]$ retains information about the membrane potential and input current at each previous time step, which implies that the dendrite layer plays a role in long-term memory. Furthermore, Eq.(6) reveals that the LM-H neuron extracts two types of information, as depicted in Fig.2. One type is historical information obtained from previous time-steps, while the other type corresponds to the current information, which aligns with the behavior of a vanilla LIF neuron. Among them, $\mu_D^l$ determines the overall proportion of historical information extraction, while $\mu_S^l$ describes the ratio of previous potential information in the historical representation. $\lambda_S^l$ affects the extent of current potential leakage, and $\lambda_D^l$ scales the input current. Notably, $\lambda_D^l$ also plays a crucial role in regulating gradient calculation in deep residual structure. Regarding the possible gradient failure on the identity path mentioned above, the corresponding calculated gradient chain for the LM-H model becomes $\prod_{l=1}^{L} \lambda_D^l H'(\lambda_D^l \boldsymbol{s}^{l-1}[t] - \theta^l)$, which means that appropriately chosen values for $\lambda_D^l$ can ensure proper gradient calculations in deep networks.

Furthermore, from Eq.(6), we note that the current LM-H model can only obtain historical information through the form of exponential decay by regulating $\mu_D^l$, which cannot flexibly determines the proportion of information extraction at each time-step. Therefore, as shown in Fig.2, we also consider a more radical version where we set membrane-related parameters at each time-step

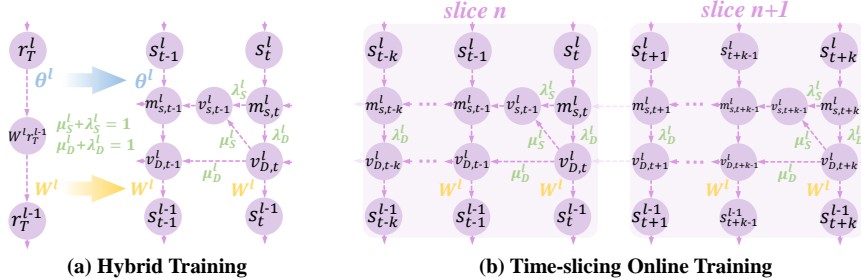

(a) Hybrid Training  (b) Time-slicing Online Training

Figure 4: Efficient Training framework for the LM-H Neuron.

$(\forall t, \mu_{D,t}^l, \mu_{S,t}^l, \lambda_{S,t}^l, \lambda_{D,t}^l)$ to be learnable, resulting in Eq.(6) becoming the following form:

$$\boldsymbol{m}_S^l[t] = \sum_{k=1}^{t-1} \lambda_{D,k}^l \prod_{j=k+1}^{t} \mu_{D,j}^l (\mu_{S,k}^l \boldsymbol{v}_S^l[k-1] + \boldsymbol{I}^l[k]) + \left( (\lambda_{S,t}^l + \lambda_{D,t}^l \mu_{S,t}^l) \boldsymbol{v}_S^l[t-1] + \lambda_{D,t}^l \boldsymbol{I}^l[t] \right). \quad (7)$$

### 4.3 A Progressive STBP Training Framework for LM-H Model

From Eq.(6), we observe that when $\mu_D^l = \mu_S^l = 0$ and $\lambda_D^l = 1$, the LM-H model degenerates into a vanilla LIF model with a decay constant equal to $\lambda_S^l$. This indicates that the LIF model is a special case of the LM-H model. Inspired by this, we propose to progressively optimize the model parameters (e.g., $\mu_D^l, \mu_S^l, \lambda_D^l, \lambda_S^l$) during the training process, starting with the degradation of the LM-H model to the LIF model.

As illustrated in Fig.3, we set $\mu_D^l, \mu_S^l, \lambda_D^l, \lambda_S^l$ in each layer as learnable parameters. To optimize these parameters stably and smoothly, we propose a scalable-shiftable sigmoid learning function defined as $h(x) = a * \sigma(x) + b$, where $a$ controls the length of the parameter optimization interval, while $b$ represents the initial position where the parameters can be chosen, and $\sigma(x)$ denotes the sigmoid function. In our training process, we set $\mu_D^l := a_{\mu,D}^l * \sigma(x_{\mu,D}^l) + b_{\mu,D}^l$, where $x_{\mu,D}^l$ is the actual learnable tensor. To maintain consistency with the initial condition of our progressive training, we set $a_{\mu,D}^l = a_{\mu,S}^l = a_{\lambda,D}^l = a_{\lambda,S}^l = 1, b_{\mu,D}^l = b_{\mu,S}^l = -0.5, b_{\lambda,D}^l = b_{\lambda,S}^l = 0.5$ and the initial values of $x_{\mu,D}^l, x_{\mu,S}^l, x_{\lambda,D}^l, x_{\lambda,S}^l$ to 0.

Eqs.(8)-(9) outline the back-propagation chain of our progressive training framework. As shown in Fig.3, our gradient calculation involves spatial and temporal dimensions. For spatial dimension, it further involves two distinct levels: the dendrite-level and soma-level. During the training process, we set $\frac{\partial \boldsymbol{v}_S^l[t]}{\partial \boldsymbol{m}_S^l[t]} = 1$ rather than $\frac{\partial \boldsymbol{v}_S^l[t]}{\partial \boldsymbol{m}_S^l[t]} = 1 - \theta^l \frac{\partial \boldsymbol{s}^l[t]}{\partial \boldsymbol{m}_S^l[t]}$. This choice allows the connection strength of various propagation chains to be entirely regulated by the parameters $\mu_D^l, \mu_S^l, \lambda_D^l, \lambda_S^l$.

$$\frac{\partial \mathcal{L}}{\partial \boldsymbol{m}_S^l[t]} = \frac{\partial \mathcal{L}}{\partial \boldsymbol{s}^l[t]} \frac{\partial \boldsymbol{s}^l[t]}{\partial \boldsymbol{m}_S^l[t]} + \frac{\partial \mathcal{L}}{\partial \boldsymbol{m}_S^l[t+1]} \frac{\partial \boldsymbol{m}_S^l[t+1]}{\partial \boldsymbol{m}_S^l[t]},$$

$$\frac{\partial \mathcal{L}}{\partial \boldsymbol{v}_D^l[t]} = \frac{\partial \mathcal{L}}{\partial \boldsymbol{m}_S^l[t]} \frac{\partial \boldsymbol{m}_S^l[t]}{\partial \boldsymbol{v}_D^l[t]} + \frac{\partial \mathcal{L}}{\partial \boldsymbol{v}_D^l[t+1]} \frac{\partial \boldsymbol{v}_D^l[t+1]}{\partial \boldsymbol{v}_D^l[t]}. \quad (8)$$

$$\frac{\partial \boldsymbol{m}_S^l[t+1]}{\partial \boldsymbol{m}_S^l[t]} = \frac{\partial \boldsymbol{m}_S^l[t+1]}{\partial \boldsymbol{v}_S^l[t]} \frac{\partial \boldsymbol{v}_S^l[t]}{\partial \boldsymbol{m}_S^l[t]} + \frac{\partial \boldsymbol{m}_S^l[t+1]}{\partial \boldsymbol{v}_D^l[t+1]} \frac{\partial \boldsymbol{v}_D^l[t+1]}{\partial \boldsymbol{m}_S^l[t]} = \lambda_S^l + \lambda_D^l \mu_S^l,$$

$$\frac{\partial \boldsymbol{v}_D^l[t+1]}{\partial \boldsymbol{v}_D^l[t]} = \frac{\partial \boldsymbol{v}_D^l[t+1]}{\partial \boldsymbol{v}_D^l[t]} + \frac{\partial \boldsymbol{v}_D^l[t+1]}{\partial \boldsymbol{m}_S^l[t]} \frac{\partial \boldsymbol{m}_S^l[t]}{\partial \boldsymbol{v}_D^l[t]} = \mu_D^l + \lambda_D^l \mu_S^l. \quad (9)$$

### 4.4 Optimizing the LM-H Model through Efficient Training

Based on previous discussions, we have established that the LIF model is a special case of the LM-H model. In this section, we aim to extend the hybrid training method proposed in (Rathi & Roy, 2023) and the online training method proposed in (Xiao et al., 2022) to the LM-H model.

Table 1: Comparison with previous SOTA works. * denotes an improved network structure.

| Dataset | Method | Architecture | Time-steps | Accuracy(%) |
|---|---|---|---|---|
| CIFAR-10 | STBP-tdBN (Zheng et al., 2021) | ResNet-19 | 4 | 92.92 |
| | Dspike (Li et al., 2021) | ResNet-18 | 4 | 93.66 |
| | TET (Deng et al., 2022) | ResNet-19 | 4 | 94.44 |
| | GLIF (Yao et al., 2022) | ResNet-18 | 4, 6 | 94.67, 94.88 |
| | | ResNet-19 | 4, 6 | 94.85, 95.03 |
| | **Ours** | **ResNet-18** | **4** | **95.62** |
| | | **ResNet-19** | **4** | **96.36** |
| CIFAR-100 | Dspike (Li et al., 2021) | ResNet-18 | 4 | 73.35 |
| | TET (Deng et al., 2022) | ResNet-19 | 4 | 74.47 |
| | GLIF (Yao et al., 2022) | ResNet-18 | 4, 6 | 76.42, 77.28 |
| | | ResNet-19 | 4, 6 | 77.05, 77.35 |
| | TEBN (Duan et al., 2022) | ResNet-19* | 4, 6 | 78.71, 78.76 |
| | **Ours** | **ResNet-18** | **4** | **78.58** |
| | | **ResNet-19** | **4** | **80.31** |
| | | **ResNet-19*** | **4** | **81.65** |
| ImageNet-200 | DCT (Garg et al., 2020) | VGG-13 | 125 | 56.90 |
| | Online-LTL (Yang et al., 2022) | VGG-13 | 16 | 54.82 |
| | Offline-LTL (Yang et al., 2022) | VGG-13 | 16 | 55.37 |
| | ASGL (Wang et al., 2023) | VGG-13 | 4, 8 | 56.57, 56.81 |
| | **Ours** | **VGG-13** | **4** | **59.93** |
| DVS-CIFAR10 | STBP-tdBN (Zheng et al., 2021) | ResNet-19 | 10 | 67.80 |
| | RecDis-SNN (Guo et al., 2022a) | ResNet-19 | 10 | 72.42 |
| | MPBN (Guo et al., 2023) | ResNet-19 | 10 | 74.40 |
| | **Ours** | **ResNet-19** | **10** | **79.10** |

**Hybrid Training.** Similar to the LIF model, the average spike firing rate of the LM-H model also follows an approximate linear relationship between layers. By combining the dynamic equations for $v_S^l[t]$ and $v_D^l[t]$ from Eq.(4), and then performing sum and average operations along the time dimension, we arrive at the following equation:

$$\frac{\sum_{t=1}^{T} s^l[t]\theta^l}{T} = \frac{\sum_{t=1}^{T} W^l s^{l-1}[t]\theta^{l-1}}{T} - \left( \frac{\sum_{t=1}^{T} \delta_D^l[t]}{T} + \frac{\sum_{t=1}^{T} \delta_S^l[t]}{T} \right),$$

$$\delta_D^l[t] = (1 - \lambda_D^l)v_D^l[t] - \mu_D^l v_D^l[t-1], \quad \delta_S^l[t] = v_S^l[t] - (\lambda_S^l + \mu_S^l)v_S^l[t-1]. \quad (10)$$

Here $T$ denotes the total time period. $\delta_D^l[t]$ and $\delta_S^l[t]$ are the conversion errors generated by the dendrite and soma layer. When $\mu_D^l + \lambda_D^l = 1, \mu_S^l + \lambda_S^l = 1$, the impact of conversion errors is further reduced. By defining $r^l[T] = \sum_{t=1}^{T} s^l[t]\theta^l/T$, we obtain the following simplified results.

$$r^l[T] = W^l r^{l-1}[T] - \left( \frac{(1 - \lambda_D^l)v_D^l[T] - \mu_D^l v_D^l[0]}{T} + \frac{v_S^l[T] - v_S^l[0]}{T} \right). \quad (11)$$

One can find that Eq.(11) further simplifies to $r^l[T] = W^l r^{l-1}[T] - \frac{v_S^l[T] - v_S^l[0]}{T}$ when $\lambda_D^l = 1, \mu_D^l = 0$. In this case, the relationship between the spike firing rates of adjacent layers in the LM-H model aligns completely with that of the LIF model. To leverage this similarity, as depicted in Fig.4(a), we initially utilize the conventional ANN-SNN conversion framework (Bu et al., 2022) to obtain a specialized LM-H model with $\lambda_D^l = 1$ and $\mu_D^l = 0$. We then proceed to optimize the model parameters through the progressive learning approach mentioned earlier. Notably, the progressive learning stage requires only a small number of epochs (typically $\leq 30$) to significantly enhance the performance of the LM-H model. Considering that the time and memory costs of ANN-SNN conversion are $O(1)$, and the corresponding costs of STBP are $O(T)$, our proposed hybrid training method will significantly save energy consumption.

**Online Training.** For the LIF model, previous works achieved gradient renewal at each step by canceling the dependencies between propagation chains at different time-steps (*e.g.* $\partial m^l[t]/\partial m^l[t-1]$), ensuring that memory consumption does not increase linearly with the growth of time-steps. However, for the LM-H model, it is not possible to cancel the gradient calculation for the learnable parameters connecting propagation chains from different time steps (*e.g.* $\mu_D^l, \mu_S^l, \lambda_S^l$). To address it, we propose an online training framework called time-slicing training. As shown in Fig.4(b), we group consecutive

Table 2: Performance of hybrid training for LM-H model.

| Dataset | Method | Architecture | Time-steps | Accuracy(%) |
|---------|--------|--------------|------------|-------------|
| CIFAR-10 | QCFS (Bu et al., 2022) | ResNet-18 | 4 | 93.66 |
| | | ResNet-20 | 4 | 83.75 |
| | **Ours** | **ResNet-18** | **4** | **94.02** |
| | | **ResNet-20** | **4** | **87.56** |
| CIFAR-100 | QCFS (Bu et al., 2022) | VGG-16 | 4 | 69.62 |
| | | ResNet-20 | 4 | 34.14 |
| | **Ours** | **VGG-16** | **4** | **73.11** |
| | | **ResNet-20** | **4** | **57.12** |
| ImageNet-200 | QCFS (Bu et al., 2022) | VGG-13 | 4 | 45.15 |
| | **Ours** | **VGG-13** | **4** | **49.09** |

Table 3: Performance of time-slicing online training on ResNet-18.

| Dataset | Method | Time-steps | Accuracy(%) |
|---------|--------|------------|-------------|
| CIFAR-10 | SLTT (Meng et al., 2023) | 6 | 94.44 |
| | **3 time-steps per slice, 2 slices** | **4, 6, 8** | **95.05, 95.42, 95.49** |
| CIFAR-100 | SLTT (Meng et al., 2023) | 6 | 74.38 |
| | 2 time-steps per slice, 2 slices | 4, 6, 8 | 76.27, 77.10, 77.56 |
| | 2 time-steps per slice, 3 slices | 4, 6, 8 | 75.99, 77.35, 77.81 |
| | 2 time-steps per slice, 4 slices | 4, 6, 8 | 74.81, 76.28, 77.01 |
| | **3 time-steps per slice, 2 slices** | **4, 6, 8** | **77.28, 78.21, 78.66** |
| | 4 time-steps per slice, 2 slices | 4, 6, 8 | 77.23, 78.30, 78.59 |

$k$ time steps into a single slice, effectively dividing the total time period into $\lceil T/k \rceil$ slices. Within each slice, we cancel the gradient calculation between different slices. After each time slice ends, a gradient refresh will be performed to ensure that the total memory overhead does not exceed $O(k)$.

## 5 EXPERIMENTS

### 5.1 EXPERIMENTAL SETTINGS

To validate the effectiveness of our proposed model and training frameworks, we conduct experimentation on various datasets, including CIFAR Family (Krizhevsky et al., 2009), ImageNet-200 (Deng et al., 2009), and DVS-CIFAR10 (Li et al., 2017) datasets. We specifically select mainstream network architectures that are consistent with previous works, including VGG (Simonyan & Zisserman, 2014) and ResNet (He et al., 2016). More additional experiments and details are provided in the Appendix.

### 5.2 COMPARISON WITH PREVIOUS STATE-OF-THE-ART WORKS

**CIFAR Family (standard dataset).** We have selected previous state-of-the-art works in various areas of SNN learning as benchmarks for comparing our proposed method. These benchmarks include optimization of learning function (Deng et al., 2022), surrogate gradient (Li et al., 2021), batchnorm layers (Zheng et al., 2021; Duan et al., 2022), and membrane-related parameters (Yao et al., 2022). As shown in Tab.1, the LM-H model demonstrates significant advantages over other methods in terms of performance. For example, on the CIFAR-100 dataset, the LM-H model achieves a top-1 accuracy of 78.58% and 80.31% on ResNet-18 and ResNet-19 with 4 time-steps, respectively, which is 2.16% and 3.26% higher than GLIF on the corresponding network architecture. Furthermore, for ResNet-19 with the improved structure, our method outperforms TEBN with a 2.94% accuracy improvement under 4 time-steps. It is worth noting that the performance of our method at 4 time-steps significantly surpasses that of other works at 6 time-steps for the CIFAR-10 and CIFAR-100 datasets.

**ImageNet-200 (large-scale dataset).** From Tab.1, we can note that the performance of our progressive training is 3.36% higher than ASGL, a current advanced neuron model with adaptive smoothing gradient. Additionally, our model exhibits superior performance at 4 time-steps compared to other methods (Garg et al., 2020; Yang et al., 2022; Wang et al., 2023) at higher time-step ranges of 8-125.

**DVS-CIFAR10 (neuromorphic dataset).** We also evaluate the representation ability of the LM-H model on the neuromorphic dataset. Compared to STBP-tdBN (Zheng et al., 2021), RecDis-

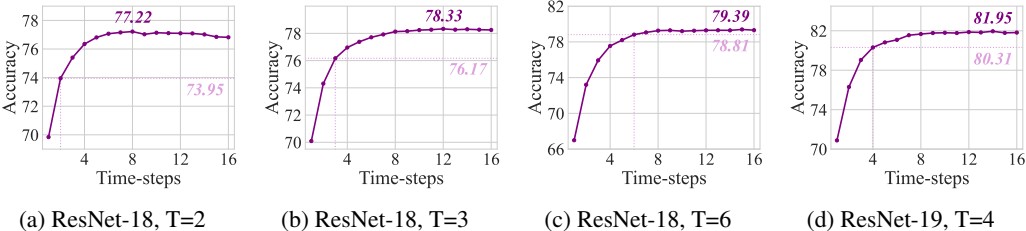

|                     |                     |                     |                     |
| :-----------------: | :-----------------: | :-----------------: | :-----------------: |
| (a) ResNet-18, T=2  | (b) ResNet-18, T=3  | (c) ResNet-18, T=6  | (d) ResNet-19, T=4  |

Figure 5: Performance of LM-H neuron (which is obtained after T-steps training) at different time-steps on the CIFAR-100 dataset.

SNN (Guo et al., 2022a) and MPBN (Guo et al., 2023), our model achieves accuracy improvements of 11.30%, 6.68% and 4.70%, which demonstrate the effectiveness of our proposed method.

### 5.3 PERFORMANCE ANALYSIS OF EFFICIENT TRAINING FRAMEWORK

**Hybrid Training.** We utilize the QCFS (Bu et al., 2022) as our ANN-SNN conversion framework during the first stage to obtain an LM-H model under special conditions ($\mu_D^l = 0, \lambda_D^l = 1, \mu_S^l + \lambda_S^l = 1$). For the progressive training stage, since we will merely optimize the LM-H model with very limited iterations (30 epochs), we can consider this stage as a fine-tuning process. For the CIFAR Family and ImageNet-200 datasets, we set the initial learning rate as $2 \times 10^{-4}$ and $1 \times 10^{-4}$, respectively. As shown in Tab.2, we note that the second stage still plays a crucial role, even with a finite number of optimization iterations. This is particularly evident for QCFS conversion models that exhibit poor performance under low time-latency conditions. For example, on the CIFAR-100 dataset, we achieve accuracy improvements of 3.49% for VGG-16 and 22.98% for ResNet-20 during further optimization within 30 epochs.

**Online Training.** We have chosen the current state-of-the-art work of online training (Meng et al., 2023) as the target for our comparison. As shown in Tab.3, our proposed scheme (3 time-steps per slice, 2 slices) outperforms SLTT by 3.83% on the CIFAR-100 dataset with 6 time-steps. In addition, we investigate the impact of slice length and slice quantity on network performance. It is evident that using a very short slice length (*e.g.* 2 time-steps per slice) may lead to a slight decrease in training precision. On the other hand, increasing the number of slices excessively (*e.g.* 4 slices) does not yield further improvements in network performance.

### 5.4 LONG-RANGE PERFORMANCE REWARD FOR LM-H MODEL

We also discover an interesting phenomenon regarding the LM-H model: despite using specific training steps during the learning process, the model's precision can still be further improved as the number of time-step increases during the inference stage. As shown in Fig.5, we use pink dotted lines to indicate the accuracy achieved under specific training steps, while purple numbers represent the peak performance of our model during the inference process. Across various training parameter configurations, we observe that the models exhibit an improvement range of approximately 0.6% to 3.3% over an extended period of time. We named this phenomenon as "Long-Range Performance Reward", which indicates that the membrane-related parameters we have learned during the training stage (*e.g.* $\mu_D^l, \mu_S^l, \lambda_S^l, \lambda_D^l$) can still effectively extract information over a prolonged time range.

## 6 CONCLUSIONS

In this paper, we propose the LM-H neuron that can comprehensively utilize historical and current information. We demonstrate that the LM-H model is the generalization of LIF neurons and addresses the calculation deficiencies associated with LIF neurons. Moreover, we design a progressive STBP training framework to dynamically regulate historical and current representation, and propose efficient training schemes to significantly reduce the computational overhead. Considering that our method has outperformed previous state-of-the-art works in multiple experiments, we believe that the proposal of the LM-H model will further promote relevant research in the field of advanced spiking neurons.

ACKNOWLEDGEMENTS

This work was supported by the National Natural Science Foundation of China (62176003, 62088102) and by Beijing Nova Program (20230484362).

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

# A   APPENDIX

## A.1   EXPERIMENTAL IMPLEMENTATION DETAILS

In this paper, we choose Stochastic Gradient Descent (SGD) optimizer (Bottou, 2012) and Cosine Annealing scheduler (Loshchilov & Hutter, 2017) for all experimental cases. Following the approach of previous works, we incorporate data-augmentation techniques to enhance the performance of SNN models, including Cutout (DeVries & Taylor, 2017) and Auto-Augment (Cubuk et al., 2019). For the DVS-CIFAR10 dataset, we resize the neuromorphic data to $48 \times 48$ pixels and integrate each image into 10 frames. In the progressive STBP training framework, we consider two optimization configurations: TET function (Deng et al., 2022) with vanilla batchnorm layers and Cross-Entropy function based on firing rate with tdBN layers (Zheng et al., 2021). We select the configuration that achieves optimal performance on the CIFAR and ImageNet-200 datasets. In addition, for ImageNet-1k dataset, we choose the combination of Cross-Entropy function, tdBN layers and an improved version of ResNet (Hu et al., 2021). For the hybrid training framework, our further training is based on the pre-trained QCFS models (Bu et al., 2022) with $L = 4$ or $L = 8$, where $L$ denotes the quantization level of the QCFS function. For time-slicing online training, we adopt the TET function to optimize gradients for each time slice. The detailed hyper-parameter settings can be found in Tab.S1.

Table S1: Experimental hyper-parameter configuration.

| Framework | Dataset | Architecture | Optimizer (lr, wd) | Batchsize | Epochs |
|---|---|---|---|---|---|
| Progressive Training | CIFAR-10 | ResNet-18 ResNet-19 | SGD $(0.025, 5 \times 10^{-4})$ | 64 | 300 |
| | CIFAR-100 | ResNet-18 ResNet-19 ResNet-19* | | | |
| | ImageNet-200 | VGG-13 | | | |
| | DVS-CIFAR10 | ResNet-19 | SGD $(0.1, 5 \times 10^{-4})$ | 32 | |
| | ImageNet-1k | ResNet-34 | SGD $(0.1, 0)$ | $64 \times 4$ | 320 |
| Hybrid Training | CIFAR-10 | ResNet-18 ResNet-20 VGG-16 | SGD $(0.002, 5 \times 10^{-4})$ | 64 | 30 |
| | CIFAR-100 | ResNet-20 | | | |
| | ImageNet-200 | VGG-13 | SGD $(0.001, 5 \times 10^{-4})$ | | |
| 2 time-steps per slice | CIFAR-10 CIFAR-100 | | SGD $(0.0125, 5 \times 10^{-4})$ | | |
| 3 time-steps per slice | CIFAR-10 CIFAR-100 | ResNet-18 | SGD $(0.01875, 5 \times 10^{-4})$ | 64 | 300 |
| 4 time-steps per slice | CIFAR-10 CIFAR-100 | | SGD $(0.025, 5 \times 10^{-4})$ | | |

## A.2   PERFORMANCE OF THE LM-H MODEL UNDER SPARSE LEARNING

We further explore the representation ability of the LM-H model under sparse weights by employing the STDS pruning strategy (Chen et al., 2022) to sparsify the weights of SNN models. The specific equations for weight sparsification are described as follows:

$$W_{\text{sparse}}^l = \text{sign}(W^l) \cdot \max(|W^l| - d, 0), \tag{S1}$$

$$d^t = \frac{1}{2}\left(\sin\left(\frac{t\pi}{T_d} - \frac{\pi}{2}\right) + 1\right) D. \tag{S2}$$

Here $d, d^t$ represents the clipping width of the current weights, while $D$ denotes the final target width. $T_d$ is the total number of updates to $d$ during the training process. Throughout the sparse training procedure, $d$ continuously increases according to Eq.S2, with $D$ set to 0.05.

Tab.S2 demonstrates the superior capability of the LM-H model in sparse learning. Even when the sparsity exceeds 80%, we still outperform ESL-SNN (Shen et al., 2023) by 4.94% and 5.83% on the CIFAR-10 and CIFAR-100 datasets, respectively. Notably, the performance of the LM-H model under

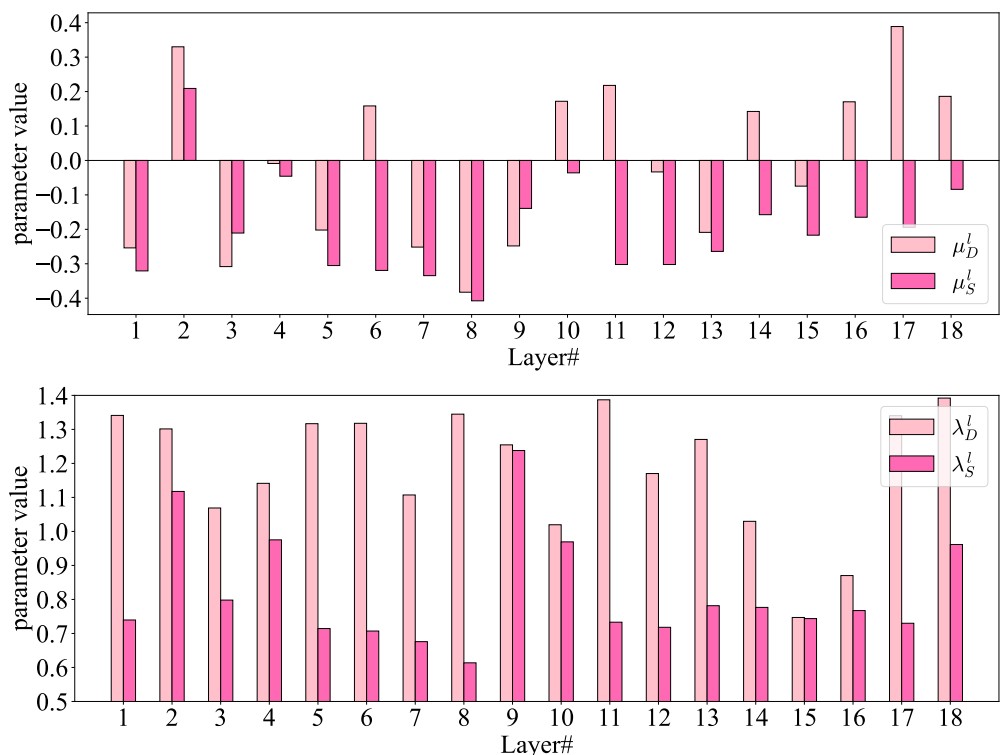

Figure S1: The learning situation for LM-H model on DVS-CIFAR10, ResNet-19.

sparse weights is significantly better than that of previous works under normal learning. In addition, we observe that despite trimming more than 80% of the weights, the LM-H model's performance does not exhibit significant degradation (with accuracy loss generally below 1%).

Table S2: Comparison of sparse STBP training with previous works.

| Dataset | Method | Architecture | Sparsity(%) | Accuracy(%) |
|---------|--------|--------------|-------------|-------------|
| CIFAR-10 | STBP-tdBN (Zheng et al., 2021) | ResNet-19 | - | 92.92 |
| | TET (Deng et al., 2022) | | - | 94.44 |
| | GLIF (Yao et al., 2022) | | - | 94.85 |
| | ESL-SNN (Shen et al., 2023) | | 50.00 | 91.09 |
| | **Ours** | **ResNet-19** | **89.59** | **96.03** |
| | | | - | 96.36 |
| CIFAR-100 | TET (Deng et al., 2022) | ResNet-19 | - | 74.47 |
| | GLIF (Yao et al., 2022) | | - | 77.05 |
| | ESL-SNN (Shen et al., 2023) | | 50.00 | 73.48 |
| | **Ours** | **ResNet-19** | **81.60** | **79.31** |
| | | | - | 80.31 |

## A.3 LEARNING SITUATION OF MEMBRANE-RELATED PARAMETERS

We visualize the learning results of the relevant membrane-related parameters on the LM-H model. As shown in Fig.S1, $\mu_D^l, \mu_S^l, \lambda_S^l, \lambda_D^l$ dynamically learn suitable values for different layers, achieving effective global information extraction layer by layer.

A.4 DETAILED EXPLANATION ABOUT EFFICIENT TRAINING FOR THE LM-H MODEL

**Hybrid Training.** The first stage of the hybrid training framework is an ANN-SNN Conversion with Quantization-Clip-Floor-Shift (QCFS) function, which can be described as follows:

$$a^l = \frac{\theta^l}{L}\text{clip}\left(\left\lfloor \frac{a^{l-1}L}{\theta^l} + \frac{1}{2} \right\rfloor, 0, L\right). \tag{S3}$$

Here $a^l$ represents the activation output of the QCFS ANN model in the $l$-th layer and $L$ denotes the quantization level. Bu et al. (2022) has proved that the discrepancy between $r^l[T]$ and $a^l$ will be optimized to a minimum value for IF model under the condition of adopting QCFS function. As the LM-H model will degenerate to a vanilla IF model when $\mu_D^l = 0, \mu_S^l + \lambda_S^l = 1, \lambda_D^l = 1$, we can obtain an LM-H model under special conditions through QCFS ANN-SNN Conversion.

Once the ANN-SNN Conversion procedure is completed, we will copy the weights and membrane-related parameters from the QCFS ANN model to the corresponding positions in the LM-H model. Due to the fact that the dendrite layer is actually hidden when $\mu_D^l = 0$, the initial value of the membrane potential $\theta^l/2$ and the learnable firing threshold $\theta^l$ are directly copied to the soma layer.

During the progressive training stage, we introduce learnable parameters for the membrane-related factors and weights to facilitate more stable fine-tuning. However, the firing thresholds are treated as scalars. As the QCFS function simulates the average firing rate of an LM-H neuron over $L$ steps, we typically set the training step $T$ at this stage to $L$. The overall algorithm description can be found in Algorithm 1.

---

**Algorithm 1** Hybrid training framework for the LM-H model.

---

**Require:** Pretrained QCFS ANN model $f_{\text{ANN}}(\boldsymbol{W}, \theta)$ with $L$ layers; Dataset $D$; Number of time-steps used for training $T$; Loss function $\mathcal{L}$; Cross-Entropy function $\mathcal{L}_{CE}$.
**Ensure:** SNN model $f_{\text{SNN}}(\boldsymbol{W}, \theta, \boldsymbol{v}_D, \boldsymbol{v}_S, \boldsymbol{s}, \mu_D, \mu_S, \lambda_S, \lambda_D)$.
1: # Convert ANN to SNN
2: **for** $l = 1$ to $L$ **do**
3:     $f_{\text{SNN}}.\boldsymbol{W}^l = f_{\text{ANN}}.\boldsymbol{W}^l$
4:     $f_{\text{SNN}}.\theta^l = f_{\text{ANN}}.\theta^l$
5:     $f_{\text{SNN}}.\boldsymbol{v}_D^l[0] = 0, f_{\text{SNN}}.\boldsymbol{v}_S^l[0] = \frac{1}{2}f_{\text{SNN}}.\theta^l$
6:     $f_{\text{SNN}}.\mu_D^l = 0, f_{\text{SNN}}.\mu_S^l + f_{\text{SNN}}.\lambda_S^l = 1, f_{\text{SNN}}.\lambda_D^l = 1$
7: **end for**
8: # Progressive training for LM-H model
9: # Set $\mu_D, \mu_S, \lambda_S, \lambda_D$ as learnable parameters and $\theta$ as scalars
10: **for** (**Image**,**Label**) in $D$ **do**
11:     # LM-H model performs forward propagation based on Eq.(4)
12:     **for** $l = 1$ to $L$ **do**
13:       **if** Use tdBN layer **then**
14:         $\{f_{\text{SNN}}.\boldsymbol{s}^l[1], ..., f_{\text{SNN}}.\boldsymbol{s}^l[T]\} = f_{\text{SNN}}(\{f_{\text{SNN}}.\boldsymbol{W}^l f_{\text{SNN}}.\boldsymbol{s}^{l-1}[1], ..., f_{\text{SNN}}.\boldsymbol{W}^l f_{\text{SNN}}.\boldsymbol{s}^{l-1}[T]\})$
15:       **end if**
16:       **if** Use vanilla batchnorm layer **then**
17:         **for** $t = 1$ to $T$ **do**
18:           $f_{\text{SNN}}.\boldsymbol{s}^l[t] = f_{\text{SNN}}(f_{\text{SNN}}.\boldsymbol{W}^l f_{\text{SNN}}.\boldsymbol{s}^{l-1}[t])$
19:         **end for**
20:       **end if**
21:     **end for**
22:     **if** Use TET loss function **then**
23:       $\mathcal{L} = \frac{1}{T}\sum_{t=1}^{T}\mathcal{L}_{CE}(f_{\text{SNN}}.\boldsymbol{s}^l[t], \textbf{Label})$
24:     **end if**
25:     **if** Use vanilla loss function **then**
26:       $\mathcal{L} = \mathcal{L}_{CE}(\frac{1}{T}\sum_{t=1}^{T}f_{\text{SNN}}.\boldsymbol{s}^l[t], \textbf{Label})$
27:     **end if**
28:     # LM-H model performs back-propagation based on Eq.(8)
29: **end for**
30: **return** $f_{\text{SNN}}(\boldsymbol{W}, \theta, \boldsymbol{v}_D, \boldsymbol{v}_S, \boldsymbol{s}, \mu_S, \mu_D, \lambda_S, \lambda_D)$

---

**Time-slicing Online Training.** Algorithm 2 has depicted the overall procedure of time-slicing training. Assume two adjacent $k$ time-steps slices: slice $n$ and slice $n + 1$. Within the slice, we perform normal progressive STBP training for the LM-H model. For the calculation chains between slices (*e.g.* $\frac{\partial \boldsymbol{m}_S^l[n \times k+1]}{\partial \boldsymbol{m}_S^l[n \times k]}$, $\frac{\partial \boldsymbol{v}_D^l[n \times k+1]}{\partial \boldsymbol{v}_D^l[n \times k]}$), we separate them from the entire gradient calculation graph, which means that forward propagation can calculate normally on these chains but back-propagation cannot. During back-propagation, the slices are completely detached from each other, which achieves gradient updates slice by slice and effectively saves computational memory overhead.

---

**Algorithm 2** Time-slicing online training framework for the LM-H model.

---

**Require:** Dataset $D$; Time-slice containing $k$ time-steps; Number of time-steps used for training $T$; Loss function $\mathcal{L}$; Cross-Entropy function $\mathcal{L}_{CE}$.
**Ensure:** SNN model $f_{\text{SNN}}(\boldsymbol{W}, \theta, \boldsymbol{v}_D, \boldsymbol{v}_S, \boldsymbol{s}, \mu_D, \mu_S, \lambda_S, \lambda_D)$.
1: # Set $\mu_D, \mu_S, \lambda_S, \lambda_D$ as learnable parameters and $\theta$ as scalars
2: **for** (**Image**,**Label**) in $D$ **do**
3:    **for** slice = 1 to $\lceil T/k \rceil$ **do**
4:       **for** $l = 1$ to $L$ **do**
5:          # LM-H model performs forward propagation based on Eq.(4)
6:          **if** Use tdBN layer **then**
7:             $\{f_{\text{SNN}}.\boldsymbol{s}^l[k \times (\text{slice}-1)+1], ..., f_{\text{SNN}}.\boldsymbol{s}^l[k \times \text{slice}]\} = f_{\text{SNN}}(\{f_{\text{SNN}}.\boldsymbol{W}^l f_{\text{SNN}}.\boldsymbol{s}^{l-1}[k \times (\text{slice}-1)+1], ..., f_{\text{SNN}}.\boldsymbol{W}^l f_{\text{SNN}}.\boldsymbol{s}^{l-1}[k \times \text{slice}]\})$
8:          **end if**
9:          **if** Use vanilla batchnorm layer **then**
10:            **for** $t = 1$ to $k$ **do**
11:               $f_{\text{SNN}}.\boldsymbol{s}^l[k \times (\text{slice}-1)+t] = f_{\text{SNN}}(f_{\text{SNN}}.\boldsymbol{W}^l f_{\text{SNN}}.\boldsymbol{s}^{l-1}[k \times (\text{slice}-1)+t])$
12:            **end for**
13:          **end if**
14:          # Detach $f_{\text{SNN}}.\boldsymbol{v}_D^l[k \times \text{slice}], f_{\text{SNN}}.\boldsymbol{v}_S^l[k \times \text{slice}]$ from the gradient computational graph
15:       **end for**
16:       # LM-H model performs gradient updates for each time-slice
17:       **if** Use TET loss function **then**
18:          $\mathcal{L} = \frac{1}{k} \sum_{t=k \times (\text{slice}-1)+1}^{k \times \text{slice}} \mathcal{L}_{CE}(f_{\text{SNN}}.\boldsymbol{s}^l[t], \textbf{Label})$
19:       **end if**
20:       **if** Use vanilla loss function **then**
21:          $\mathcal{L} = \mathcal{L}_{CE}(\frac{1}{k} \sum_{t=k \times (\text{slice}-1)+1}^{k \times \text{slice}} f_{\text{SNN}}.\boldsymbol{s}^l[t], \textbf{Label})$
22:       **end if**
23:       # LM-H model performs back-propagation based on Eq.(8)
24:    **end for**
25: **end for**
26: **return** $f_{\text{SNN}}(\boldsymbol{W}, \theta, \boldsymbol{v}_D, \boldsymbol{v}_S, \boldsymbol{s}, \mu_S, \mu_D, \lambda_S, \lambda_D)$

---

Table S3: Experimental results about the radical version on multiple datasets.

| Dataset | Method | Architecture | Time-steps | Accuracy(%) |
|---|---|---|---|---|
| CIFAR-10 | Ours | ResNet-18 | 4 | 95.62 |
| | **Ours (radical version)** | **ResNet-18** | **4** | **95.82** |
| CIFAR-100 | Ours | ResNet-18 | 4 | 78.58 |
| | **Ours (radical version)** | **ResNet-18** | **4** | **78.90** |
| ImageNet-200 | Ours | VGG-13 | 4 | 59.93 |
| | **Ours (radical version)** | **VGG-13** | **4** | **60.37** |
| ImageNet-1k | STBP-tdBN (Zheng et al., 2021) | ResNet-34 | 6 | 63.72 |
| | TET (Deng et al., 2022) | ResNet-34 | 6 | 64.79 |
| | RecDis-SNN (Guo et al., 2022a) | ResNet-34 | 6 | 67.33 |
| | MBPN (Guo et al., 2023) | ResNet-34 | 4 | 64.71 |
| | SEW ResNet (Fang et al., 2021a) | ResNet-34 | 4 | 67.04 |
| | GLIF (Yao et al., 2022) | ResNet-34 | 4 | 67.52 |
| | **Ours (radical version)** | **ResNet-34** | **4** | **69.73** |

### A.5 COMPARISON BETWEEN THE VANILLA AND RADICAL VERSIONS OF THE LM-H MODEL

As shown in Eq.(6), the overall proportion term for historical information is represented as $(\mu_D^l)^{t-k}$, indicating that the extracted proportions at different time-steps follow an exponential distribution. In contrast, in Eq.(7), the corresponding term changes to $\prod_{j=k+1}^{t} \mu_{D,j}^l$, allowing for more freedom and randomness in the extracted proportions. In addition, in the radical version, relevant factors that regulate the membrane potential ($\mu_{S,t}^l, \lambda_{S,t}^l$) and input current ($\lambda_{D,t}^l$) are independently assigned at each time-step, making the information representation range of the radical version wider than that of the vanilla version.

Tab.S3 shows the relevant performance of the radical version on CIFAR and ImageNet datasets, which demonstrates that the radical version does have the advantage of learning membrane-related parameters more precisely. In addition, during the experimental procedure, we noticed that the convergence speed of the radical version of the training accuracy was also ahead of the vanilla version, especially on large-scale datasets.

### A.6 ADDITIONAL EXPERIMENTS ON VGG STRUCTURE

We have also conducted additional experiments using the VGG architecture on the CIFAR-10 and CIFAR-100 datasets. As shown in Tab.S4, our results demonstrate superior performance compared to previous SOTA methods, even when employing smaller network structures and fewer time-steps.

Table S4: Comparison with previous works on the CIFAR datasets.

| Dataset | Method | Architecture | Time-steps | Accuracy(%) |
|---|---|---|---|---|
| CIFAR-10 | Diet-SNN (Rathi & Roy, 2023) | VGG-16 | 5 | 92.70 |
| | Real Spike (Guo et al., 2022b) | VGG-16 | 5, 10 | 92.90, 93.58 |
| | **Ours** | **VGG-13** | **4** | **93.98** |
| | **Ours (radical version)** | **VGG-13** | **4** | **94.80** |
| CIFAR-100 | Diet-SNN (Rathi & Roy, 2023) | VGG-16 | 5 | 69.67 |
| | RecDis-SNN (Guo et al., 2022a) | VGG-16 | 5 | 69.88 |
| | Real Spike (Guo et al., 2022b) | VGG-16 | 5, 10 | 70.62, 71.17 |
| | **Ours** | **VGG-13** | **4** | **73.99** |
| | **Ours (radical version)** | **VGG-13** | **4** | **74.79** |

