# OpenReview forum: "A Progressive Training Framework for Spiking Neural Networks with Learnable Multi-hierarchical Model"
_ICLR.cc/2024/Conference — ICLR 2024 poster_

### Official Review · Reviewer_LLZy · 2023-10-29

**Soundness:** 3 good
**Presentation:** 3 good
**Contribution:** 4 excellent
**Rating:** 8
**Confidence:** 5

**Summary:**

This paper analyzes the limitations of the vanilla LIF neuron model, including the problems of gradient vanishing and exploding, as well as the inability to differentiate the current response by extracting past information. Based on these analyses, the authors propose a novel learnable multi-hierarchical model that has a wider calculation scope along the time dimension by incorporating the functions of dendrite and soma. Additionaly, they further design a progressive STBP training framework for the LM-H model.

**Strengths:**

1.The proposed model is neural-inspired.

2.The work is solid. The authors provide rigorous theoretical analysis of the LM-H model and demonstrate that the LIF model is merely a subset of the LM-H model.

3.The progressive training method efficiently solves the multi-parameter learning problem of the LM-H model.

4.Experimental results demonstrate the significant advantages of the LM-H model across multiple datasets.

**Weaknesses:**

1.The authors illustrate the issues of gradient vanishing and exploding faced by the LIF neuron. However, they have not demonstrated how the proposed LM-H model addresses these problems.

2.The figures displayed in this paper solely depict the vanilla LM-H model. It would be good if the authors can also integrate the radical version of the LM-H model into these figures.

**Questions:**

1. Please provide a more comprehensive explanation and analysis of the radical version of the LM-H model, and discuss its advantages compared to the vanilla LM-H model.

2. Regarding the hybrid training framework of the LM-H model, the author used the conversion framework based on the IF model during the ANN-SNN conversion phase. How about LIF neurons? Does it fit into this framework?

3. The authors have illustrated the distinction between their work and that of TC-LIF in the related works section. I suggest adding it into the introduction as well.


-------------------------------------------
Thank you for the clarification and for incorporating my suggestions in the revised version! Having read all the reviews, I concur with the opinions of the other reviewers that the authors have made a substantial contribution to developing novel neuron models for SNNs. This work will be of sufficient significance to advance the field of neuromorphic computing.

---

> ### Author Response · Authors · 2023-11-16
> **To Reviewer LLZy**
>
> ## To Reviewer LLZy
> > How the proposed LM-H model addresses the issues of gradient vanishing and exploding faced by the LIF neuron?
>
> Thanks for your comments! We have provided a thorough analysis and a detailed explanation for this concern. Please refer to **To All Reviewers**.
>
> > It would be good if the authors can also integrate the radical version of the LM-H model into these figures.
>
> Thanks for your suggestion! We have revised the figure and submitted a new version.
>
> > Please provide a more comprehensive explanation and analysis of the radical version of the LM-H model, and discuss its advantages compared to the vanilla LM-H model.
>
> As shown in Eq.6, the overall proportion term for historical information is represented as $(\mu_D)^{t-k}$, indicating that the extracted proportions at different time-steps follow an exponential distribution. In contrast, in Eq.7, the corresponding term changes to $\prod_{j=k+1}^{t} \mu_{D,j}$, allowing for more freedom and randomness in the extracted proportions. In addition, in the radical version, relevant factors that regulate the membrane potential ($\mu_{S,t}, \lambda_{S,t}$) and input current ($\lambda_{D,t}$) are independently assigned at each time-step, making the information representation range of the radical version wider than that of the vanilla version. We have added more details of the comparsion in Section A.5 of the revised version.
>
>
> > The author used the conversion framework based on the IF model during the ANN-SNN conversion phase. How about LIF neurons? Does it fit into this framwork?
>
> Yes, it also fits into this framwork. According to our proposed hybrid training framework, if we initially convert a pre-trained ANN to the IF model, then set the membrane leaky constant $\lambda^l$ in each layer as a learnable parameter and adopt STBP for training, we can also incorporate the LIF model into the same framework. However, as shown in Eq.6, the LIF model can be regarded as a special case of the LM-H model, with its representation range being limited to a subset of the LM-H model's representation range. Consequently, we are inclined to believe that the significance of hybrid training based on the LIF model is quite limited.
>
>
> > The authors have illustrated the distinction between their work and that of TC-LIF in the related works section. I suggest adding it into the introduction as well.
>
> Thanks for pointing it out. We have added the relevant content in the introduction of our revised version.

---

> > ### Author Response · Authors · 2023-11-22
> >
> > Dear Reviewer LLZy:
> >
> > We want to express our gratitude for dedicating your valuable time to review our paper meticulously and provide us with your valuable insights and feedbacks.
> >
> > As the Author-Review Discussion period is approaching the end, we would like to ensure that we have addressed all the concerns you raised. If you have any unresolved issue, we are ready to offer further clarification or make necessary modifications.
> >
> > Thank you for your time and effort.
> >
> > Best regards,
> >
> > The Authors

---

### Official Review · Reviewer_T4SQ · 2023-11-05

**Soundness:** 2 fair
**Presentation:** 2 fair
**Contribution:** 2 fair
**Rating:** 6
**Confidence:** 3

**Summary:**

This work proposed a new neuron model, Learnable Multi-hierarchical (LM-H) model, to effectively extracting global information along the time dimension and propagate gradients in deep networks. In the LM-H model, the scaling factors from the dendrite layer regulate the proportion of historical information extracted by the model, while the factors from the soma layer determine the degree of potential leakage and the intensity of the input current at present.

**Strengths:**

The proposed LMH model, along with GLIF, TC-LIF, enriches the family of the spiking neuron models. This approach is unique and original, as it combines existing ideas in a new way to solve a problem in Spiking Neural Networks. The proposed model and training algorithm address the deficiencies of the widely adopted LIF model and offer a new approach to solving problems in this field.

**Weaknesses:**

While the proposed model and training algorithm are unique and original, the paper could benefit from a more detailed discussion of why they propose the new model, how they differ from existing methods, and what specific contributions they make to the field of Spiking Neural Networks.
What are the significant deficiencies in deep layer gradient calculation and capturing global information on the time dimension, as mentioned in this paper?

**Questions:**

Does the author address the gradient vanishing & exploding problem in deep residual architectures with the new LMU model?
How?
The authors listed results of cifar data set on resnet, can you provide results on vgg as well?

---

> ### Author Response · Authors · 2023-11-16
> **To Reviewer T4SQ (Part I)**
>
> ## To Reviewer T4SQ
> > What are the significant deficiencies in deep layer gradient calculation? Does the author address the gradient vanishing & exploding problem in deep residual architectures with the new LM-H model? How?
>
> Thanks for your comments! We have provided a thorough analysis and a detailed explanation for this concern. Please refer to **To All Reviewers**.
>
> > What are the significant deficiencies in capturing global information on the time dimension, as mentioned in this paper?
>
> In terms of capturing global information, the LIF model is limited compared to the LM-H model, as depicted in Eq.6. The LIF model can only consider the second term of the equation, which represents the current representation. This limitation arises from the nature of spiking neurons (LIF model), which can transmit binary information only once per time-step. If the membrane potential and spike firing information from different time-steps are not comprehensively utilized, the representation capability of the output spike sequence becomes restricted. Consequently, this limitation can have an impact on the performance of SNNs. Table 1 in this paper, along with Table R1, can demonstrate the significant performance gap between the vanilla LIF model and our LM-H model, particularly when applied to large-scale datasets.
>
>
>
> > The paper could benefit from a more detailed discussion of why they propose the new model, how they differ from existing methods, and what specific contributions they make to the field of Spiking Neural Networks.
>
> Thanks for your valuable feedback! We would like to clarify that our motivation for proposing the LM-H model primarily stems from the limitations of the existing LIF model in addressing deep gradient computation and global information extraction challenges. While other advanced models such as GLIF and TC-LIF have been proposed by researchers in the SNN community, we believe that the LM-H model offers four distinct contributions that previous works have not encompassed:
>
> (1) **The relationship between the LM-H model and the LIF model is elucidated through rigorous mathematical analysis, highlighting the superiority of the LM-H model.** We demonstrate that the LM-H model encompasses the LIF model as a special case (Eq.5-Eq.6), while also showcasing the superior information extraction capabilities of the LM-H model. This analysis ensures that the theoretical performance of the LM-H model is at least on par with the LIF model, as acknowledged by Reviewer LLZy.
>
>
> (2) **The LM-H model possesses a remarkable capability to dynamically and flexibly extract global information across different time steps**. Through detailed analysis, we identify the specific functions of membrane-related parameters and set appropriate learning ranges for them. This allows the LM-H model at the $t$-th timestep to effectively incorporate relevant information, including input current and membrane potential from previous $t-1$ steps. This capability to consider information from multiple time steps simultaneously sets the LM-H model apart from other advanced models.
>
> (3) **The LM-H model demonstrates superior scalability and transferability.** From a computational perspective, the LM-H model is a more general form of the LIF model. In terms of neuronal structure, the LM-H model incorporates a dendrite layer to store historical information for each time-step. When $\mu_D=\mu_S=0, \lambda_D=1$, the dendrite layers become invisible, and the LM-H model degenerates into a single-layer state consistent with the LIF model. These two aspects establish a close relationship between the LM-H model and the LIF model, facilitating easy migration of various training frameworks associated with the LIF model (such as hybrid training and online training) to the LM-H model.
>
>
> (4) **The progressive STBP training framework is novel.** This novel training framework starts with the parameter settings of the LIF model as the initial state and dynamically optimizes membrane-related parameters during the training process to achieve the optimal state of the LM-H model. This design concept of "from special to general" allows for efficient and effective training of the LM-H model.

---

> ### Author Response · Authors · 2023-11-16
> **To Reviewer T4SQ (Part II)**
>
> > The authors listed results of cifar data set on resnet, can you provide results on vgg as well?
>
> Thanks for your suggestion! We have conducted additional experiments using the VGG-13 architecture on the CIFAR-10 and CIFAR-100 datasets. As shown in Tab. R4, our results demonstrate superior performance compared to previous SOTA methods, even when employing smaller network structures and fewer time-steps. We have also included these content in the Appendix A.6 of our submitted version.
>
> **Table R4: Comparison with previous SOTA works on CIFAR datasets, VGG structure.**
> | Dataset     | Method | Architecture | Time-steps | Accuracy(%) |
> | ----------- | ------ | ------------ | ---------- | ----------- |
> |CIFAR-10|Diet-SNN [1]|VGG-16|5|92.70|
> |CIFAR-10|Real Spike [2]|VGG-16|5, 10|92.90, 93.58|
> |**CIFAR-10**|**Ours**|**VGG-13**|**4**|**93.98**|
> |**CIFAR-10**|**Ours(radical version)**|**VGG-13**|**4**|**94.80**|
> |CIFAR-100|Diet-SNN [1]|VGG-16|5|69.67|
> |CIFAR-100|RecDis-SNN [3]|VGG-16|5|69.88|
> |CIFAR-100|Real Spike [2]|VGG-16|5, 10|70.62, 71.17|
> |**CIFAR-100**|**Ours**|**VGG-13**|**4**|**73.99**|
> |**CIFAR-100**|**Ours(radical version)**|**VGG-13**|**4**|**74.79**|
>
> [1] Nitin Rathi and Kaushik Roy. DIET-SNN: A low-latency spiking neural network with direct input encoding and leakage and threshold optimization. IEEE Transactions on Neural Networks and Learning Systems, pp. 1–9, 2021.
>
> [2] Yufei Guo, Liwen Zhang, Yuanpei Chen, Xinyi Tong, Xiaode Liu, YingLei Wang, Xuhui Huang, Zhe Ma. Real spike: Learning real-valued spikes for spiking neural networks. European Conference on Computer Vision. 2022.
>
> [3] Yufei Guo, Xinyi Tong, Yuanpei Chen, Liwen Zhang, Xiaode Liu, Zhe Ma, and Xuhui Huang. RecDis-SNN: Rectifying membrane potential distribution for directly training spiking neural networks. In IEEE Conference on Computer Vision and Pattern Recognition, pp. 326–335, 2022.

---

> > ### Comment · Reviewer_T4SQ · 2023-11-19
> >
> > Upon revisiting the manuscript and carefully considering the author's responses, I acknowledge the efforts made to address my concerns. The clarifications provided have significantly improved my understanding of certain aspects of the work. As a result, I am willing to adjust my initial score from 5 to 6.

---

> > > ### Author Response · Authors · 2023-11-22
> > >
> > > Dear Reviewer T4SQ:
> > >
> > > We want to express our gratitude for dedicating your valuable time to review our paper meticulously and provide us with your valuable insights and feedbacks.
> > >
> > > Thank you for your time and effort.
> > >
> > > Best regards,
> > >
> > > The Authors

---

### Official Review · Reviewer_uzHG · 2023-11-09

**Soundness:** 3 good
**Presentation:** 3 good
**Contribution:** 2 fair
**Rating:** 6
**Confidence:** 3

**Summary:**

The authors of this paper address the limitations of the widely used LIF model in SNN by proposing a novel LM-H model. The LM-H model overcomes issues related to gradient calculation in deep networks and capturing global information along the time dimension. The authors also develop a progressive training algorithm specifically for the LM-H model. Experiments on various datasets demonstrate the superior performance of the LM-H model and the training algorithm compared to previous state-of-the-art SNN approaches.

**Strengths:**

1 Did a lot baselines and achieved SOTA performance
2 A novel neuron model LH-M that is a extension of LIF model

**Weaknesses:**

1. In the context of deep residual architectures, LIF neurons are known to exhibit issues to either vanishing or exploding gradients. How has the LH-M model effectively addressed and mitigated these challenges?

2. Both ImageNet and CIFAR datasets focus on static image classification, whereas the LH-M model incorporates historical data and demonstrates superior performance. Could you elucidate the underlying reasons for this enhanced performance in the context of LH-M's utilization of historical information?

3. How was the conversion from Artificial Neural Networks (ANN) to Spiking Neural Networks (SNN) executed, given that ANNs do not inherently incorporate temporal information? Considering the significance of historical data within the LH-M model, how was this temporal aspect effectively integrated during the conversion process?

**Questions:**

1. LIF neurons have a gradient vanishing or exploding problem in deep residue architecture, how did LH-M solved this problem?

2. ImageNet and CIFAR are all static image classification datasets, but LH-M involves historical information and performed better. Can you give an explaination?

3. How did you perform ANN2SNN conversion? ANN do not include temporal information, but in LH-M historical information is an important property.

---

> ### Author Response · Authors · 2023-11-16
> **To Reviewer uzHG**
>
> ## To Reviewer uzHG
> > LIF neurons have a gradient vanishing or exploding problem in deep residual architecture, how did LM-H solved this problem?
>
> Thanks for your comments! We have provided a thorough analysis and a detailed explanation for this concern. Please refer to **To All Reviewers**.
>
>
> > ImageNet and CIFAR are all static image classification datasets, but LH-M involves historical information and performed better. Could you elucidate the underlying reasons for this enhanced performance in the context of LH-M's utilization of historical information?
>
> Thanks for your insightful comment! While ImageNet and CIFAR are static datasets, our experimental process, as consistent with previous works, involves inputting each image for multiple time-steps. Since spiking neurons (model) can only emit binary spikes (either firing or not firing) at each time-step, the representation range of information transmitted at a single time-step is inherently limited. Therefore, in order to enhance the representation ability of spiking neurons, it becomes necessary to consider the membrane potential ($\forall j\in[1, t-1], \boldsymbol{v}^l[j]$) and spike firing status ($\forall j\in[1, t-1], \boldsymbol{s}^l[j]$) at each previous time-step when deciding whether to fire a spike at the current time-step. This utilization of historical information allows for a comprehensive decision-making process. In fact, regardless of the input data type (static or neuromorphic), as long as we adopt multi-step training, the LM-H model can observe a broader range of information compared to the LIF model, thereby demonstrating its superior performance, as illustrated in Eq.6 in the main text.
>
>
> > How did you perform ANN2SNN conversion? ANN do not include temporal information, but in LM-H historical information is an important property.
>
> Thanks for pointing it out. We would like to clarify that our hybrid learning method can be divided into two stages. In the first stage, we adopt traditional ANN-SNN Conversion framework to obtain a LM-H model under the special case, which indeed does not contain temporal information. As shown in Eq.11 and its corresponding analysis, we derived that the most accurate mathematical mapping relationship between the converted LM-H model and the pre-trained ANN model (i.e., the minimum conversion error) is established under the condition $\mu_D=0,\lambda_D=1,\mu_S+\lambda_S=1$. Therefore, in the first stage of hybrid training, we replace the activation functions in the pre-trained ANN model with the LM-H models, adhering to the special case ($\mu_D=0,\lambda_D=1,\mu_S+\lambda_S=1$) layer by layer.
>
> However, in the second stage of the training process, all the membrane-related parameters of the LM-H model ($\mu_D,\lambda_D,\mu_S,\lambda_S$) are set as learnable parameters. Throughout the progressive training process, these parameters are dynamically optimized and updated to effectively extract temporal information. The detailed description and specific procedure of the algorithm can be found in Appendix A.4 of the revised version, where a comprehensive explanation is provided.

---

> > ### Author Response · Authors · 2023-11-22
> >
> > Dear Reviewer uzHG:
> >
> > We want to express our gratitude for dedicating your valuable time to review our paper meticulously and provide us with your valuable insights and feedbacks.
> >
> > As the Author-Review Discussion period is approaching the end, we would like to ensure that we have addressed all the concerns you raised. If you have any unresolved issue, we are ready to offer further clarification or make necessary modifications.
> >
> > Thank you for your time and effort.
> >
> > Best regards,
> >
> > The Authors

---

> ### Author Response · Authors · 2023-11-23
> **Thank you for the time and look forward to your feedback.**
>
> Dear Reviewer uzHG,
>
> As the Author-Review Discussion period will be closed within a few hours, we would like to briefly summarize your concerns and our relevant responses as follow:
> 1. The first concern is about the gradient calculation problem in deep residual architecture. We have provided a thorough analysis to discuss the specific difference of the gradient calculation between the vanilla LIF and LM-H model, as shown in To All Reviewers.
> 2. The second concern is about the LH-M model's utilization of historical information on static datasets. We have made a detailed explanation to point out that the LM-H model can observe a broader range of information compared to the LIF model as long as we adopt multi-step training, thereby demonstrating its superior performance.
> 3. The third question is about the utilization of temporal information in our hybrid training framework. We have clarified that our hybrid learning method can be divided into two stages. Throughout the progressive training stage, the membrane-related parameters are dynamically optimized and updated to effectively extract temporal information, the detailed description can be found in Appendix A.4 of our revised version.
>
> Based on these facts and positive feedback from other reviewers, we sincerely hope that you can reconsider your initial rating. If you have any further comment, please let us know and we are glad to address your concern.
>
> Best regards,
>
> The Authors

---

> ### Comment · Reviewer_uzHG · 2023-11-23
> **Thank you for your comment**
>
> I appreciate the authors' thorough and well-structured rebuttal, which has addressed my concerns effectively. It is evident that they have made solid progress in refining their work and providing convincing arguments to support their claims. Additionally, I believe that authors proposed a novel and effective spiking neuron model for SNN community. Taking these revisions into account, I am inclined to raise the paper's score.

---

### Official Review · Reviewer_SiwN · 2023-11-10

**Soundness:** 2 fair
**Presentation:** 3 good
**Contribution:** 3 good
**Rating:** 5
**Confidence:** 3

**Summary:**

The paper extends the LIF model for SNN to a more generalized version called LM-H, enhances its flexibility by making certain parameters learnable, and designs a progressive learning procedure to effectively train the network. Some experiments on relatively small datasets were presented to show that the proposed approach is superior to relevant prior methods.

**Strengths:**

The LM-H model and the learning algorithm were respectively inspired by and similar to existing works, however put together the paper still proposed a novel and practical framework for SNN learning.

**Weaknesses:**

The experiments only covered several relatively small datasets. The ImageNet 200 dataset was the largest one in the paper is actually a tiny subset of ImageNet. Datasets like ImageNet 1k/22k would be more convincing to validate the practical value of the proposed approach.

Performance comparison in the experiment section is somewhat inconsistent, e.g. only ImageNet 200 results included the radical version showing better performance.

Grammar errors scatter through the paper, further proof-reading is suggested.

**Questions:**

Why only small datasets were experimented upon, was it because the proposed approach has scalability issues on larger and more practical datasets?

Why the performance of the radical version was only presented for ImageNet while missing for other datasets, was it because that it didn't show better accuracy?

---

> ### Author Response · Authors · 2023-11-16
> **To Reviewer SiwN**
>
> ## To Reviewer SiwN
> > Why only small datasets were experimented upon, was it because the proposed approach has scalability issues on larger and more practical datasets?
>
> Thank you for your suggestion! We have conducted additional experiments on the ImageNet-1k dataset and compared the performance of our model with previous SOTA approaches using the ResNet-34 architecture. Due to the time constraints during the rebuttal period, we directly evaluated the performance of our radical version. As illustrated in Tab. R2, it is evident that our method outperforms other SOTA approaches by at least 2% with the same (or fewer) time-steps, which clearly demonstrates the superior scalability of the LM-H model on large-scale datasets. Due to page constraints, we have included these content in the Appendix of the revised paper (Tab. S3 of Section A.5). In the final version, we will incorporate it into the main text.
>
>
> **Table R2: Comparison with previous SOTA works on ImageNet-1k dataset.**
> | Dataset     | Method | Architecture | Time-steps | Accuracy(%) |
> | ----------- | ------ | ------------ | ---------- | ----------- |
> |ImageNet-1k|STBP-tdBN [1]|ResNet-34|6|63.72|
> |ImageNet-1k|TET [2]|ResNet-34|6|64.79|
> |ImageNet-1k|RecDis-SNN [3]|ResNet-34|6|67.33|
> |ImageNet-1k|MBPN [4]|ResNet-34|4|64.71|
> |ImageNet-1k|SEW ResNet [5]|ResNet-34|4|67.04|
> |ImageNet-1k|GLIF [6]|ResNet-34|4|67.52|
> |**ImageNet-1k**|**Ours**|**ResNet-34**|**4**|**69.73**|
>
> > Why the performance of the radical version was only presented for ImageNet while missing for other datasets, was it because that it didn't show better accuracy?
>
> Thanks for pointing it out. We have expanded our comparative experiments to include both the vanilla version and the radical version on CIFAR-10 and CIFAR-100 datasets, as shown in Tab. R3.  Given the inherent capability of the residual structure to enhance accuracy, particularly on relatively simple datasets, we have also incorporated the commonly used VGG structure to facilitate a clearer comparison between the two versions. As depicted in Table R3, the radical version, by regulating gradient calculation and information extraction, continues to exhibit better performance than the vanilla version on the CIFAR-10 and CIFAR-100 datasets. Furthermore, during the experimental process, we observed that the radical version also displayed faster convergence in terms of training accuracy, particularly when dealing with large-scale datasets. Due to page constraints, we have included these content in the Appendix of the revised paper (Tab.S3 of Section A.5, Tab.S4 of Section A.6). In the final version, we will incorporate it into the main text.
>
>
> **Table R3: Comparison of vanilla and radical versions on CIFAR-10/100 datasets.**
> | Dataset     | Method | Architecture | Time-steps | Accuracy(%) |
> | ----------- | ------ | ------------ | ---------- | ----------- |
> |CIFAR-10|vanilla version|VGG-13|4|93.98|
> |CIFAR-10|**radical version**|**VGG-13**|**4**|**94.80**|
> |CIFAR-10|vanilla version|ResNet-18|4|95.62|
> |CIFAR-10|**radical version**|**ResNet-18**|**4**|**95.82**|
> |CIFAR-100|vanilla version|VGG-13|4|73.99|
> |CIFAR-100|**radical version**|**VGG-13**|**4**|**74.79**|
> |CIFAR-100|vanilla version|ResNet-18|4|78.58|
> |CIFAR-100|**radical version**|**ResNet-18**|**4**|**78.90**|
>
>
> > Grammar errors scatter through the paper, further proof-reading is suggested.
>
> Thanks for your advice！We have carefully corrected the grammar errors in this paper and uploaded a new version.
>
> [1] Hanle Zheng, Yujie Wu, Lei Deng, Yifan Hu, and Guoqi Li. Going deeper with directly-trained larger spiking neural networks. In AAAI Conference on Artificial Intelligence, pp. 11062–11070, 2021.
>
> [2] Shikuang Deng, Yuhang Li, Shanghang Zhang, and Shi Gu. Temporal efficient training of spiking neural network via gradient re-weighting. International Conference on Learning Representations, 2022.
>
> [3] Yufei Guo, Xinyi Tong, Yuanpei Chen, Liwen Zhang, Xiaode Liu, Zhe Ma, and Xuhui Huang. RecDis-SNN: Rectifying membrane potential distribution for directly training spiking neural networks. In IEEE Conference on Computer Vision and Pattern Recognition, pp. 326–335, 2022.
>
> [4] Yufei Guo, Yuhan Zhang, Yuanpei Chen, Weihang Peng, Xiaode Liu, Liwen Zhang, Xuhui Huang, and Zhe Ma. Membrane potential batch normalization for spiking neural networks. In Proceedings of the IEEE/CVF International Conference on Computer Vision, 2023.
>
> [5] Wei Fang, Zhaofei Yu, Yanqi Chen, Tiejun Huang, Timothée Masquelier, and Yonghong Tian. Deep residual learning in spiking neural networks. In Advances in Neural Information Processing Systems. 2022.
>
> [6] Xingting Yao, Fanrong Li, Zitao Mo, and Jian Cheng. GLIF: A unified gated leaky integrate-and-fire neuron for spiking neural networks. In Advances in Neural Information Processing Systems, 2022.

---

> > ### Author Response · Authors · 2023-11-22
> >
> > Dear Reviewer SiwN:
> >
> > We want to express our gratitude for dedicating your valuable time to review our paper meticulously and provide us with your valuable insights and feedbacks.
> >
> > As the Author-Review Discussion period is approaching the end, we would like to ensure that we have addressed all the concerns you raised. If you have any unresolved issue, we are ready to offer further clarification or make necessary modifications.
> >
> > Thank you for your time and effort.
> >
> > Best regards,
> >
> > The Authors

---

> ### Author Response · Authors · 2023-11-23
> **Thank you for the time and look forward to your feedback.**
>
> Dear Reviewer SiwN,
>
> As the Author-Review Discussion period will be closed within a few hours, we would like to briefly summarize your concerns and our relevant responses as follow:
> 1. The first concern is about the scalability of our proposed method on larger datasets. We have conducted additional experiments on the ImageNet-1k dataset and compared the performance of our model with previous SOTA approaches. The experimental results in Tab.R2 have demonstrated the superior scalability of the LM-H model on large-scale datasets.
> 2. The second concern is about the comparison between the vanilla version and the radical version. We have expanded our comparative experiments in Tab.R3 to include both the vanilla version and the radical version on CIFAR-10 and CIFAR-100 datasets. The experimental results can verify the effectiveness of the radical version.
> 3. The third question is about the further proof-reading of this paper. We have carefully corrected the grammar errors in this paper and uploaded a new version.
>
> Based on these facts and positive feedback from other reviewers, we sincerely hope that you can reconsider your initial rating. If you have any further comment, please let us know and we are glad to address your concern.
>
> Best regards,
>
> The Authors

---

### Author Response · Authors · 2023-11-16
**To All Reviewers**

## To All Reviewers
**We sincerely thank all reviewers for their insightful and constructive feedback!** We are encouraged that they found our paper "novel" [Reviewer SiwN, uzHG, LLZy], "practical" [Reviewer SiwN], "unique" [Reviewer T4SQ], "original" [Reviewer T4SQ] and "solid" [Reviewer LLZy], as well as "providing rigorous theoretical analysis" [Reviewer LLZy] and "enriching the family of the spiking neuron models" [Reviewer T4SQ]. We are pleased that they found our method has "superior performance" [Reviewer uzHG] and "significant advantages" [Reviewer LLZy] across multiple datasets. We will carefully answer the relevant questions raised by every reviewer. We sincerely hope that you can make brand-new evaluations for our work.

Here we would like to address the general concerns surrounding the issues of gradient vanishing and exploding, as raised by most reviewers.


> What are the significant deficiencies about gradient calculation in deep residual architecture? Does the author address the gradient vanishing & exploding problem with the new LM-H model? How?

Thanks for this constructive question! Yes, our LM-H model can solve the gradient vanishing & exploding problem of the LIF model in deep residual architecture.

In previous studies [1], it has been noted that for the Spiking ResNet with LIF neurons, the overall gradient can be expressed as $\frac{\partial s^L[t] }{\partial s^0[t]}=\prod_{l=1}^L H'({s}^{l-1}[t]-\theta^l)$, given that the identity mapping condition is satisfied (refer to Section 4.1 of our paper). Here, $s^l[t]$ represents the firing status of neurons in layer $l$ at time-step $t$, $\theta^l$ is the firing threshold of neurons in layer $l$, and $H(\cdot)$ denotes the Heaviside step function. However, as the network depth $L$ increases, this formulation can lead to the problem of gradient vanishing or exploding. This is due to the fact that ${s}^{l-1}[t]$ is a binary variable, and $H'({s}^{l-1}[t]-\theta^l)$ for each layer can be either greater than 1 or less than 1.

In contrast, for the LM-H model, the corresponding formulation becomes $\frac{\partial s^L[t] }{\partial s^0[t]}=\prod_{l=1}^L \lambda_D^l H'(\lambda_D^l {s}^{l-1}[t]-\theta^l)$ or $\frac{\partial s^L[t] }{\partial s^0[t]}=\prod_{l=1}^L \lambda_{D,t}^l H'(\lambda_{D,t}^l {s}^{l-1}[t]-\theta^l)$ (for the radical version). This means that each layer has a learnable parameter $\lambda_D^l$ (or $\lambda_{D,t}^l$ for the radical version, where each time-step has a different learnable parameter) that can dynamically adjust the gradients. Consequently, this approach effectively addresses the issue of gradient vanishing or exploding (see page 5 of the revised paper for details).

The experimental results provide further validation. We conducted a comparison between the Spiking ResNet with LIF neural model and the LM-H model using the ImageNet-1k dataset. Tab. R1 show that the deeper 34-layer Spiking ResNet with LIF neurons achieves a lower test accuracy (61.86%) compared to the shallower 18-layer Spiking ResNet (62.32%), a phenomenon known as performance degradation, as indicated in reference [1]. However, by replacing the LIF neural models with the LM-H model, the 34-layer ResNet achieves a top-1 accuracy of 69.73%.

**Table R1: Comparison with the LIF and LM-H models on ImageNet-1k.**
| Dataset     | Method | Architecture | Time-steps | Accuracy(%) |
| --- | ------ | ------------ | ---------- | ----------- |
| ImageNet-1k | vanilla LIF | ResNet-18 | 4 | 62.32 |
| ImageNet-1k | vanilla LIF | ResNet-34 | 4 | 61.86 |
| ImageNet-1k | vanilla LIF | ResNet-50 | 4 | 57.66 |
| ImageNet-1k | vanilla LIF | ResNet-101 | 4 | 31.79 |
|**ImageNet-1k**|**Ours**|**ResNet-34**|**4**|**69.73**|

[1] Wei Fang, Zhaofei Yu, Yanqi Chen, Tiejun Huang, Timothée Masquelier, and Yonghong Tian. Deep residual learning in spiking neural networks. In Advances in Neural Information Processing Systems. 2022.

---

### Meta-Review · Area_Chair_kiDu · 2023-12-10

**Metareview:**

This paper generated considerable discussion amongst the reviewers, and I congratulate the authors on their detailed and comprehensive rebuttals, which were critical in leading some reviewers to raise their scores. Ultimately, 3/4 reviewers scored the paper as above threshold for acceptance, and after reading their comments I am inclined to agree with this majority assessment.  I'm pleased to report that this paper has been accepted to ICLR.  Congratulations!  Please revise the manuscript to address all reviewer comments and questions.

**Justification For Why Not Higher Score:**

There was one reviewer who gave a very enthusiastic appraisal (score = 8), while the other three (5,6,6) were much closer to the threshold. Ultimately it seems to me that a poster accept is the right decision, based on the level of overall enthusiasm.

**Justification For Why Not Lower Score:**

There was one reviewer who gave a very enthusiastic appraisal (score = 8), while the other three (5,6,6) were much closer to the threshold. Ultimately it seems to me that a poster accept is the right decision, based on the level of overall enthusiasm.

---

### Decision · Program_Chairs · 2024-01-16

Accept (poster)